# Seismic Wave Attenuation Characteristics from the Ground Motion Spectral Analysis around the Kanto Basin

**Ying Zhou [1], Tianming Miao [1,\*], Jian Yang [1], Xiuli Wang [1], Hongwei Wang [2] and Wenzhong Zheng [3]**

[1] School of Civil Engineering, Jilin Jianzhu University, Changchun 130118, China; joey052zy@163.com (Y.Z.); fengshangxingyu@163.com (J.Y.); wxl374495135@163.com (X.W.)

[2] Institute of Engineering Mechanics, China Earthquake Administration, Harbin 150080, China; whw1990413@163.com

[3] School of Civil Engineering, Harbin Institute of Technology, Harbin 150090, China; zheng_wenzhong@yeah.net

\* Correspondence: tianmingmiao@yeah.net

**Abstract:** In order to study the seismic wave attenuation characteristics of complex plate tectonics in and around the Kanto Basin, based on the focal mechanism and Slab1.0 model, the research area is divided into four regions. The one-step non-parametric generalized inversion technique was used to analyze the seismic wave attenuation characteristics of each region separately. The results show that the seismic path attenuation of earthquakes occurring in the shallow crust (Reg.1) is weak, and the seismic wave refraction at the crust–mantle boundary leads to almost no attenuation over a long hypocentral distance (>60 km), the frequency–dependent inelastic attenuation is also weak with the 0.5–20 Hz quality factor $Q = 92.33f^{1.87}$. The seismic path attenuation of the upper mantle earthquakes occurring in the Kanto Basin (Reg.2) is strong, and the attenuation curve decreases with the increase of hypocentral distance, which is approximately parallel to the geometric diffusion $R^{-2.0}$, the frequency–dependent inelastic attenuation is stronger with the quality factor $Q = 27.75f^{1.08}$. The seismic path attenuation of subduction zone earthquakes (Reg.3 and Reg.4) is more obvious in the high–frequency band and has a frequency correlation, indicating that the attenuation of subduction zone earthquakes includes more inelastic attenuation. The frequency–dependent inelastic attenuation $Q$ of Reg.3 and Reg.4 are $52.58f^{0.95}$ and $58.07f^{0.89}$, respectively.

**Keywords:** earthquake ground motions; Fourier analysis; path attenuation; quality factor

## 1. Introduction

The regional geological structure of the subduction zone is complex and accompanied by frequent crust–mantle activities. The rapid release of energy in the activity leads to the subduction zone being prone to rare earthquakes, and also causes a series of secondary disasters such as tsunamis and fires. Many scholars have studied the interplate tectonics of the subduction zone by analyzing the earthquakes occurring in the subduction zone around the Pacific Ocean [1–5]. The Kanto Basin is the most densely populated area in Japan, and it is also the area with the most complex subsurface structure. At the surface is the Eurasian plate, with the Philippine plate subducting under the Eurasian plate, and the Pacific plate subducting under the Philippine plate forming a triple–layered subduction situation. The friction and torsion between the Philippine plate and the Eurasian plate, the compression of the Pacific plate as it sinks beneath the Philippine plate, and the interactions between the three plates have caused many large earthquakes [6]. The devastating $M_w = 7.9$ Kanto Earthquake that occurred on 1 September 1923 is considered a rare event that occurs every 200–400 years, it killed an estimated 140,000 people and damaged more than 370,000 buildings [7,8]. Complex geological structure and frequent seismic activity lead to high seismic risk in the Kanto Basin [9–11], which also makes it an ideal place for studying seismic attenuation characteristics of complex plate tectonics.

The ground–motion prediction equation (GMPE) plays an important role in earthquake engineering, probabilistic seismic hazard analysis, and seismic design of structures, with the GMPE empirical model often used in probabilistic seismic hazard analysis. Many GMPE models have been proposed in the recent decade, including the widely used Next Generation Attenuation model [12–15]. However, there are few studies on GMPEs in the subduction zone complex plate region [16–19], and its specific earthquake type is not taken as a model parameter of GMPEs [20,21]. Zhao et al. [22] divided the earthquake types in the subduction zone of Japan into four types and proposed three GMPEs for different earthquake types, respectively for subduction slab earthquakes [23], subduction interface earthquakes [24], shallow crust, and upper mantle earthquakes [25]. It is pointed out that the inelastic attenuation rate of ground motion is the fastest at the subduction interface and the slowest at the subduction slab. The inelastic attenuation rate of the upper mantle is higher than that of the shallow crust, but they are consistent in the long period (>2 s) [25].

In 1986, Andrews first proposed the generalized spectrum inversion method, which can separate the source spectrum, path attenuation, and site effect of ground motion in the frequency domain, and has been used in the study of regional ground motion characteristics [26–29]. However, due to the assumption of geometric diffusion, the quality factor results occasionally do not conform to actual physics. The non-parametric generalized spectrum inversion method proposed by Castro [30] can effectively solve the singular value problem of quality factors. The calculation is divided into two steps. In the first step, a discrete variable related to distance is used to represent the attenuation of seismic waves in the propagation medium. In the second step, the source spectrum and site effect are separated in the Fourier spectrum modified with attenuation [31–33]. Although the two-step method can obtain accurate attenuation, when separating attenuation in the first step, the site response is averaged into the source term, resulting in a systematic deviation between the site and the source in the second step. In order to solve this problem, Oth [34] combined the traditional method with the two-step method and proposed the one-step non-parametric generalized spectrum inversion method. Solving these three factors in one step, not only guaranteed the authenticity of attenuation but also prevented system deviation.

The intensive arrangement of the Kyoshin Network (K–NET) and Kiban Kyoshin Network (KiK–net) provide good conditions for inversion research of the strong–motion recordings in Japan. Previous studies focused on a large geographical area and did not classify the area according to different earthquake types. In this study, according to the focal mechanism and Slab1 0 model, we first divide the Kanto Basin and its surrounding areas into four regions, including a shallow crustal seismic region, an upper mantle seismic region, and two subduction zone seismic regions. Then, the one-step non-parametric generalized inversion method is used to analyze the path attenuation of each region. We obtained the attenuation results for each of the four regions and then compared and discussed the seismic wave attenuation characteristics of shallow crust, upper mantle, and subduction zone. The study of propagation path attenuations is basic for the reliable prediction of ground motions, and the assessment of seismic hazard [35,36]. Reliable estimation of attenuation is of great significance to the earthquake prevention and disaster reduction of cities and the improvement of seismic capacity.

## 2. Methodology

We apply a one-step non-parametric generalized inversion technique proposed by Oth et al. [34] in 2011 to separate the S–wave of the observed spectral amplitude $O(f, M_i, R_{ij})$ into source spectra $S(f, M_i)$, path attenuation $A(f, R_{ij})$, and site effect $G(f)$ in the frequency domain. The first step is to separate the path attenuation from the observed spectral amplitude:

$$\ln O(f, M_i, R_{ij}) = \ln S(f, M_i) + \ln G(f) + \ln A(f, R_{ij}) \qquad (1)$$

where i represents the number of events and j represents the number of stations, $R_{ij}$ is the hypocentral distance from the hypocenter of the ith event to the jth station, and f is the frequency. $M_i(f)$ is a scalar that depends on the size of the ith event which includes both the

source spectrum of the ith event and the average site response of all the records obtained by the ith event. $A(f,R_{ij})$ is a seismic attenuation function described by distance, which is used to represent the attenuation characteristics of seismic waves in the propagation medium at different frequencies, including all factors leading to attenuation of the path (e.g., geometrical spreading, anelastic, and scattering attenuation, refracted arrivals, etc.). Based on the inelastic properties of the shallow crust and the view that seismic amplitude is negatively correlated with distance, $A(f,R_{ij})$ satisfies the following two assumptions: (1) $A(f,R_{ij})$ is not supposed to have any parametric functional form and is constrained to be a smooth function of distance; (2) $A(f,R_{ij})$ is constrained to 1.0 at the reference distance $R = R_{ref}$, independent of frequency.

Expressing Equation (1) as a matrix formulation:

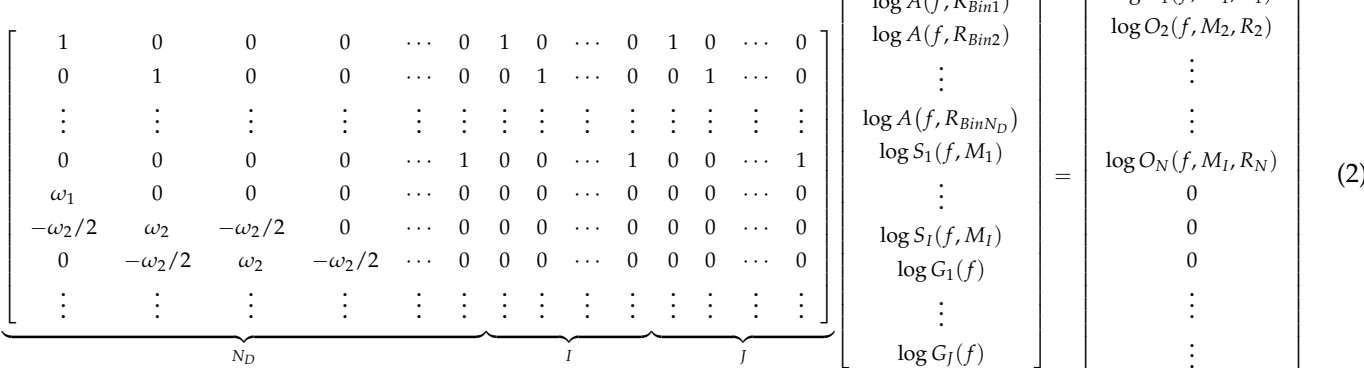

$$(2)$$

where, $\omega_1$ and $\omega_2$ are the weight coefficients, which are used to represent the constraint condition of $A(f,R_{ij})$, $\omega_1$ specifies the attenuation of $A(f,R_{ij})$ from the reference distance $R_{ref}$ and $\omega_2$ constrain the smoothing characteristics of $A(f,R_{ij})$. After several test calculations, the $\omega_1$ and $\omega_2$ used in this study are 20 and 500, respectively.

## 3. Data Set

K–NET and KiK–net are strong–motion seismograph networks covering the whole of Japan, with both of them operated by the National Research Institute for Earth Science and Disaster Resilience (NIED). Each K–NET station has a seismograph on the ground surface, while the KiK–net station has two seismographs installed in the borehole and on the corresponding ground surface.

In order to study the attenuation characteristics of the Kanto Basin and its surroundings (139–141.5° E, 35–37.5° N), the K–NET records and the KiK–net surface records obtained from earthquakes for $M_{JMA}$ 4.0–7.0 which occurred from 1 January 2010 to 31 December 2019 were analyzed by using a one-step non-parametric generalized inversion technique. A total of 881 earthquakes were released by NIED and shown in Figure 1. The circles indicate the location of the earthquake, with different colors and sizes distinguishing the source depth and magnitude. The gray area represents the land of Honshu Island, and the white area represents the sea on the east side of Honshu Island. Interestingly, we found that the focal depth distribution of the source has obvious regional characteristics. That is, for earthquakes that occurred in the Kanto Basin, the focal depth is more than 40 km, for earthquakes that occurred in the land area on the northeast of the Kanto Basin, the focal depth is generally less than 20 km. In accordance with the Crust 1.0 model, the average crust thickness in and around the Kanto Basin in Japan is about 25 km, indicating that earthquakes in the Kanto Basin mostly occur in the upper mantle, while earthquakes on the northeast of the Kanto Basin mostly occur in the shallow crust. For those earthquakes that occurred on the eastern coast of Honshu Island with focal depths that ranged from 10 km to 50 km, this suggests that they may include earthquakes that occurred in both the shallow crust and the upper mantle. According to the geographical concentration distribution of earthquakes and the regional characteristics of focal depth, we divided the research scope

into four regions. It should be noted that we only roughly divided the scope so that we need to process, analyze, and match the recordings in each region, to obtain the final data set and accurate scope in each region. The dotted lines in Figure 1 are used as a symbolic boundary to separate the whole area into four regions: Region 1 (Reg.1) is located on the northeast land of Kanto Basin, and the focal depth of earthquakes is mainly within 20 km. Region 2 (Reg.2) is located in the Kanto Basin and the focal depth of earthquakes is almost more than 40 km. Region 3 (Reg.3) and Region 4 (Reg.4) are located off the eastern coast of Honshu Island, and the focal depths mainly range from 20 km to 50 km.

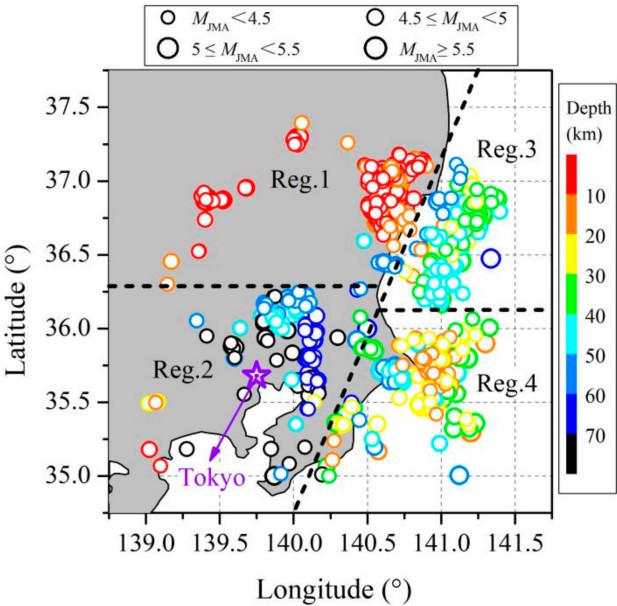

**Figure 1.** The location, magnitude, and focal depth of 881 earthquakes that occurred in the range of 139–141.5° E, 35–37.5° N from 1 January 2010 to 31 December 2019. The circles indicate the location of the earthquake, with different colors and sizes distinguishing the source depth and magnitude. The black dashed lines represent the approximate division of the source location region, which roughly divides the research scope into regions—Reg.1, Reg.2, Reg.3, and Reg.4.

The data sets for spectral inversion of the four regions are constructed according to the following criteria: (1) To reduce the contaminations from the coda wave to the extracted S wave, the hypocentral distance of the selected records are less than 100 km; (2) The peak ground acceleration (PGA) shall not be greater than 50 cm/s$^2$ to avoid the existence of nonlinear soil behavior as much as possible; (3) For each region, each selected event should be recorded by at least ten stations, each of which should collect at least ten records that match criteria (1) and (2).

After the preliminary manual screening, we processed the obtained three–component acceleration records. Each record was processed by baseline correction and a Butterworth bandpass filter. The high–cut corner frequency ($f_{hc}$) was uniformly set to 25 Hz lower than the Nyquist frequency, while the low–cut corner frequency ($f_{lc}$) was determined according to the quality of each record, such as the signal–to–noise ratio (SNR) [37,38] and the low–frequency spectral shape of the Fourier amplitude spectrum (FAS), as suggested by Goulet et al. [39]. Figure 2a shows an example of how to pick up the S–wave window and Pre–P wave noise window. The arrival of the S–wave and P–wave was identified as the abruptly increased point in the Husid [40] plot, and the end of the S–wave was calculated by the cumulative root–mean–square (RMS) function [41], see Equations (3) and (4), respectively. Concurrently, the end time of the S–wave was also identified by the energy method proposed by Pacor et al. [42]. On the premise of ensuring that the S–wave length satisfied the resolution requirements of $f_{lc}$ for each record, the smaller value of the end time identified by the cumulative RMS and the energy method should be selected. The Pre–P

wave noise window needs to be extracted with the same length as the S–wave window until the P–wave arrives. Cosine tapers were added on both ends of the extracted S–waves to eliminate truncation errors, and the length of each taper corresponded to 10% of the S–wave duration [43–45], as shown in Figure 2b.

$$H_n(t) = \frac{\int_0^T [a(t)]^2 dt}{\int_0^\infty [a(t)]^2 dt} \quad (3)$$

$$CRMS = \sqrt{\frac{1}{T} \int_0^T \|a(t)\|^2 dt} \quad (4)$$

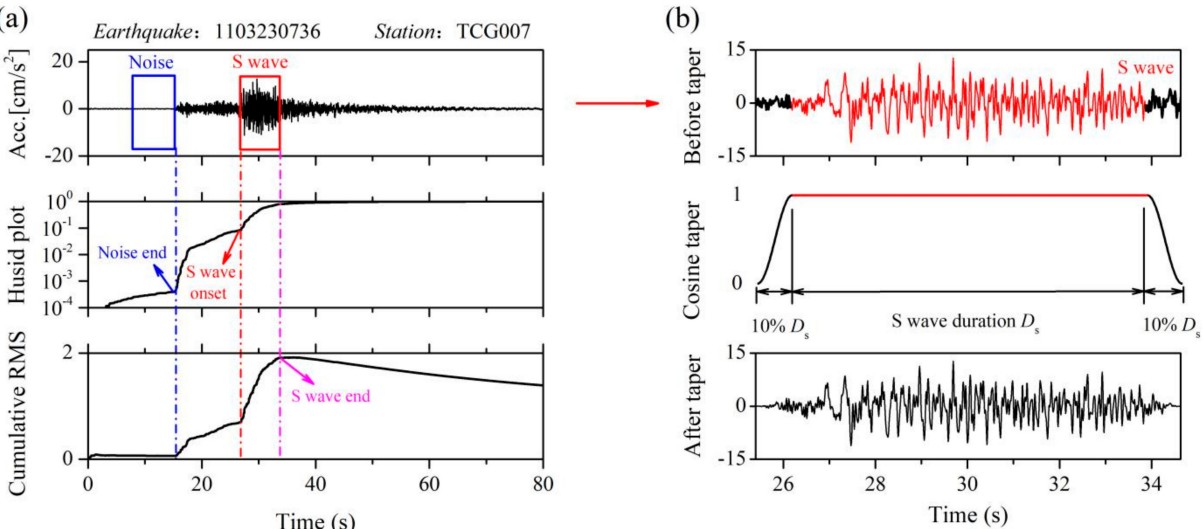

**Figure 2.** (**a**) An example illustrating how to extract the S wave signal and the pre-P wave noise window according to the Husid plot and the cumulative RMS function. (**b**) The example of how to add cosine windows at the beginning and end of the seismic wave.

The Fourier amplitude spectrum (FAS) is calculated and smoothed by using the window function of Konno and Ohamachi [46] with parameter b = 20 for the extracted S–wave and pre–P wave noise windows with same length. The SNR was used to evaluate the quality of data, so as to eliminate the influence of noise on the generalized inversion results. There is no uniform specification for the SNR threshold. Oth et al. [33,34,37], Sharma et al. [33], and Jeong et al. [28] used the SNR threshold of 3, Picozzi et al. [47] used an SNR threshold of 10. The minimum SNR applied at any FAS used in the inversion of this study is 5, to make the signal as free from background noise as far as possible. The $f_{hc}$ and $f_{lc}$ of the illustrated records in Figure 3a are 0.5 Hz and 25 Hz, and the spectra are usable at a frequency range from $1.25f_{hc}$ to $f_{lc}/1.25$ [48], as shown in the shaded area. It should be noted that if the number of frequency points of a record whose SNR $\geq$ 5 is less than 90% of the total number of frequency points (i.e., the SNR passing rate of the record is less than 90%), the whole record will be considered to be of poor quality and will not be used in the inversion. Figure 3b plots the number of usable spectra at each frequency. It was found that the obvious increase in the number of usable spectra occurs when the frequency is higher than 0.35 Hz.

Finally, a total of 12,198 strong–motion recordings obtained from 397 earthquakes were assembled in the dataset that was used for the inversion. The distribution of 397 earthquakes is shown in Figure 4, with different colored circles representing different types of earthquakes. According to the seismic focal mechanism solution automatically measured by the AQUA system of NIED, Japan, and based on the geometric structure

model of Slab1.0 global subduction zone established by Hayes et al. [49], the types of earthquakes in our study were identified according to the earthquake classification scheme adapted for Japan proposed by Zhao et al. [22]. There are four types of earthquakes (see Figure 4): shallow crustal (red circles), upper mantle (cyan circles), subduction interface (yellow circles), and subduction slab earthquake (pink circles). Some earthquakes without focal mechanism solutions are classified as unidentified earthquakes (black circles). The types and corresponding numbers of earthquakes in each region are shown in Table 1.

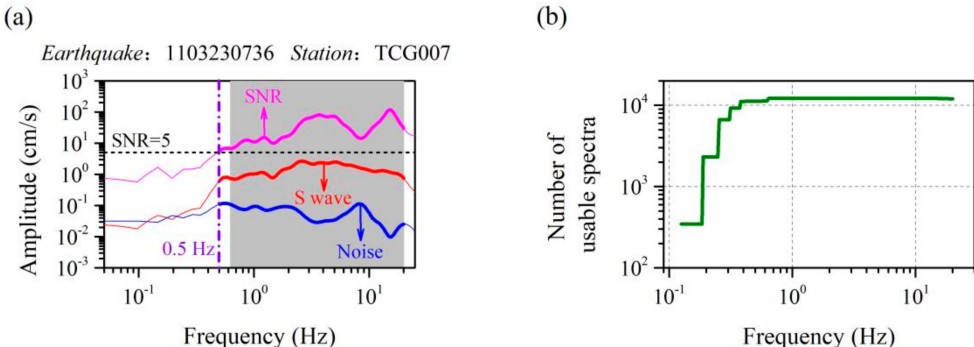

**Figure 3.** (**a**) The Fourier amplitude spectra for the extracted S wave (red solid line) and the pre-P wave noise window (blue solid line), and calculated signal-to-noise (SNR) (pink solid line). The shaded area indicates the usable frequency band determined according to the $1.25f_{hc}$ and SNR. (**b**) The number of usable spectra at each frequency.

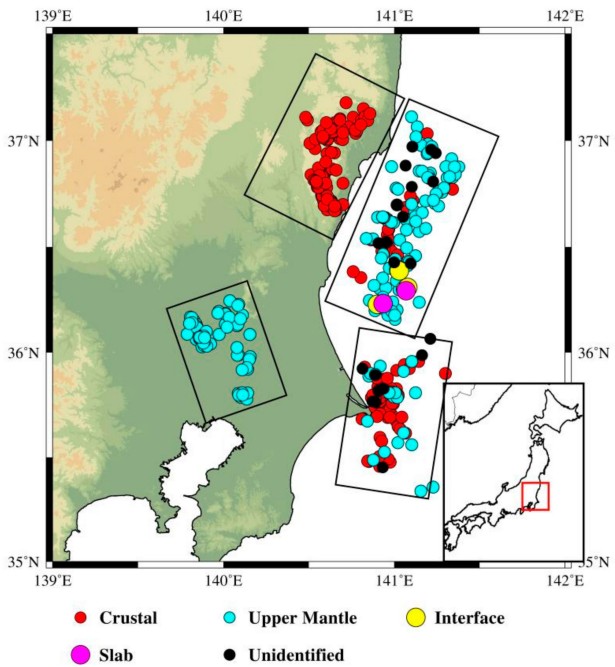

**Figure 4.** Distribution of 397 earthquakes (circles) considered in this study.

**Table 1.** Type and number of earthquakes included in each region.

| Region | Crustal | Upper Mantle | Interface | Slab | Unidentified |
|--------|---------|--------------|-----------|------|--------------|
| Reg.1  | 98      | -            | -         | -    | -            |
| Reg.2  | -       | 69           | -         | -    | -            |
| Reg.3  | 26      | 90           | 5         | 2    | 12           |
| Reg.4  | 64      | 21           | -         | -    | 9            |

According to the location and type of source in Figure 4, it can be found that the source distribution presents obvious regional characteristics. As we suspected, the earthquakes in Reg.1 all occurred in the shallow crust, and the earthquakes in Reg.2 all occurred in the upper mantle. The earthquakes in Reg.3 and Reg.4 occurred in the subduction area of the Pacific Ocean, and the complex and various types of earthquakes are the characteristics of the subduction zone. To determine the research scope of the four regions, we first determined the earthquakes contained in each region, and select the final research scope of each region according to the distribution of propagation paths between earthquakes and stations. The distribution of earthquakes and stations in each region is shown in Figure 5. The red circles represent the epicenter of earthquakes considered in this study, and the magnitude is indicated by the different scales. The green triangles represent the location of strong–motion stations, which are all located on the surface of the land in this study.

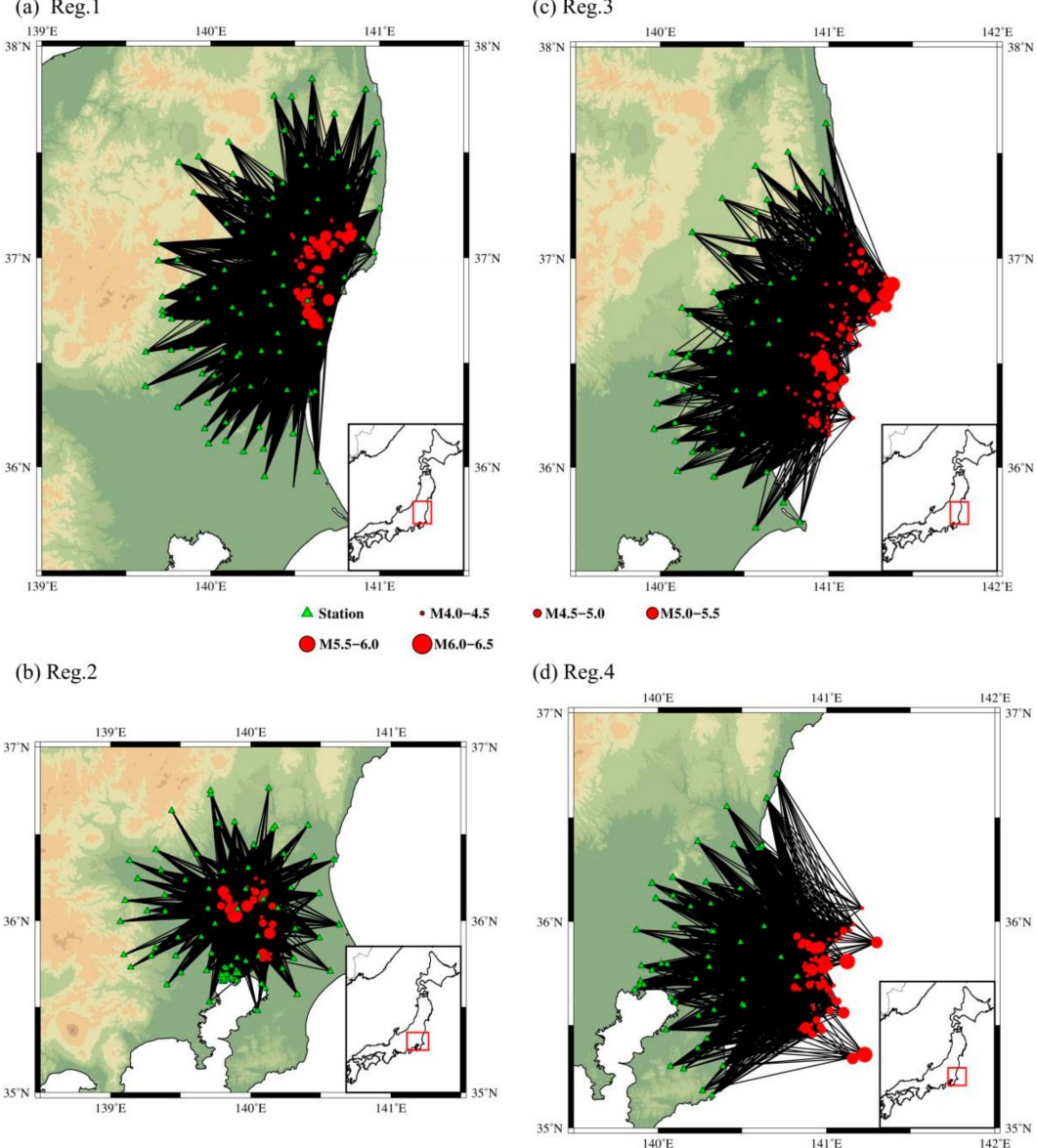

**Figure 5.** Earthquake epicentres (circles) and strong–motion stations (triangles) considered in (**a**) Reg.1, (**b**) Reg.2, (**c**) Reg.3, and (**d**) Reg.4.

Figure 6 shows the histogram of the focal depth for each region and the distribution of hypocentral distance vs. magnitude ($M_{JMA}$) of the records used for the inversion. The earthquakes in Reg.1 all occurred in the shallow crust of the land to the north of the Kanto

Basin, with $M_{JMA} < 5.5$. The hypocentral distances were mainly evenly distributed in the range of 40–100 km, with the minimum and maximum of 10.58 km and 100 km, respectively. The stations were set on the surface, and the propagation path from earthquakes to stations was all in the shallow crust within the land. Therefore, the analysis of Reg.1 was helpful to understand the path attenuation characteristics of shallow crustal earthquakes propagation in the crust. The earthquakes in Reg.2 mainly occurred in the upper mantle of the Kanto Basin with a maximum magnitude of $M_{JMA}$ 5.5. As all the focal depths were greater than 40 km, the hypocentral distances of the records were greater than 60 km. Analyzing the path attenuation characteristics of seismic waves in Reg.2 was of great significance to understanding the propagation of seismic waves in the basin. The earthquakes in Reg.3 and Reg.4 all occurred in the sea on the eastern side of the Kanto Basin. The types of earthquakes were complex and various, with obvious seismic characteristics of the subduction zone. The hypocentral distance of the earthquakes in the two regions was ~20–100 km, and both regions had earthquakes with $M_{JMA} > 6$. As shown in Figure 4, Reg.4 contained only three earthquake types including shallow crustal, upper mantle, and some unidentified, while Reg.3 included subduction interface and subduction slab in addition to the above three types, indicating that Reg.3 was more consistent with the characteristics of the subduction zone. The specific number of recordings and earthquakes, and the range of magnitude, focal depth, and hypocentral distance in each region are shown in Table 2.

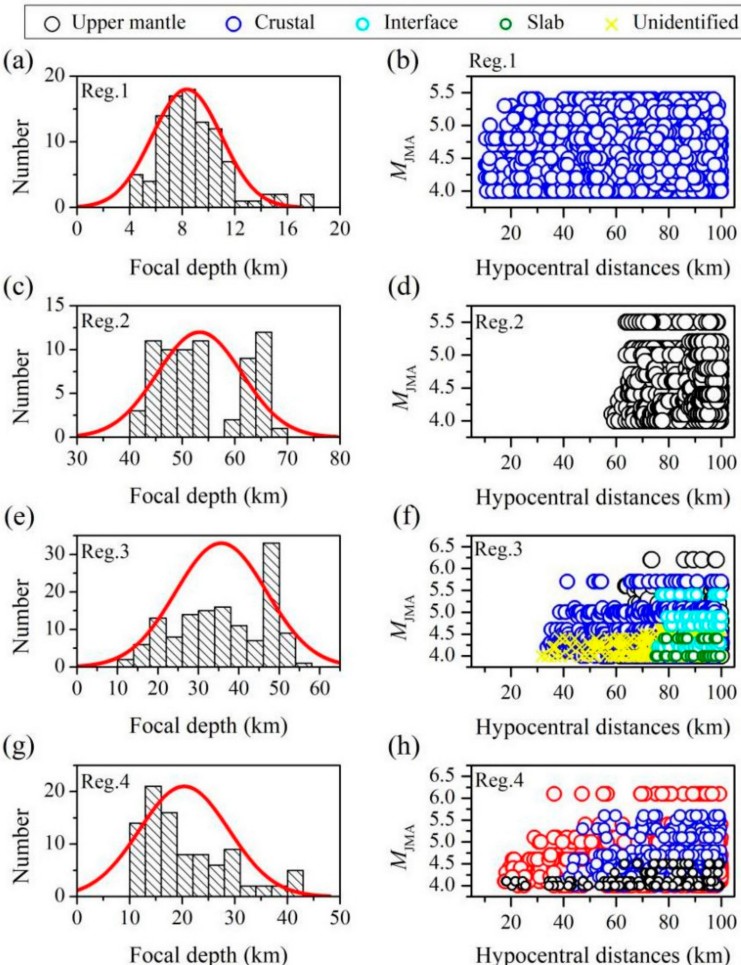

**Figure 6.** Histogram of the focal depth for (**a**) Reg.1, (**c**) Reg.2, (**e**) Reg.3, and (**g**) Reg.4. Distribution of hypocentral distance vs. magnitude of the records used for (**b**) Reg.1, (**d**) Reg.2, (**f**) Reg.3, and (**h**) Reg.4.

**Table 2.** Number of recordings and earthquakes, magnitude, focal depth, Hypocentral distance in each region.

| Region | Number of Recordings | Number of Earthquakes | Magnitude | Focal Depth | Hypocentral Distance |
| --- | --- | --- | --- | --- | --- |
| Reg.1 | 4697 | 98 | $M_{JMA}$ 4–5.4 | 4–17 km | 10–100 km |
| Reg.2 | 2990 | 69 | $M_{JMA}$ 4–5.5 | 41–67 km | 59–100 km |
| Reg.3 | 2412 | 136 | $M_{JMA}$ 4–6.2 | 13–55 km | 10–100 km |
| Reg.4 | 2109 | 94 | $M_{JMA}$ 4–6.1 | 10–43 km | 18–100 km |

## 4. Results and Discussion

The one-step non-parametric generalized inversion method is used to analyze the path attenuation in the four regions, and the results were compared with the linear geometric diffusion attenuation model, including $R^{-0.5}$, $R^{-1.0}$, $R^{-1.5}$, and $R^{-2.0}$, as shown in Figure 7. There is almost no correlation between path attenuation and frequency in Reg.1, and the path attenuation curves of each frequency (0.25–20 Hz) have little difference. In general, the path attenuation in Reg.1 is relatively weak, and the attenuation curves are roughly between the geometric diffusion $R^{-0.5}$ and $R^{-1.0}$. The attenuation curves with a hypocentral distance less than 60 km show a downward trend, and the slope is close to $R^{-0.5}$. However, when the hypocentral distance is 60 km, the slope of the attenuation curves bend roughly horizontally, resulting in less decrease in the value of the attenuation curves within the hypocentral distances of 60–100 km. The significantly weakening of attenuation at long distances (>60 km) may be related to the seismic wave refraction at the crust–mantle boundary. Since the focal depth of earthquakes in the Kanto Basin (Reg.2) is greater than 40 km, resulting in a large hypocentral distance, the curves only reflect the path attenuation within the hypocentral distances of 60–100 km. Unlike the far–field path attenuation of Reg.1, the path attenuation of Reg.2 is stronger over a long distance. The path attenuation curves mainly range between geometric diffusion $R^{-1.0}$ and $R^{-1.5}$, and the slope is approximately $R^{-2.0}$. The attenuation curves have a significant negative correlation with hypocentral distance, but no correlation with frequency. The path attenuation of Reg.2 is stronger than that of Reg.1, mainly because the seismic waves generated by the earthquake in Reg.2 pass through the upper mantle and crustal layers when received by the surface station, and its propagation path is more complex and longer than that of Reg.1. The propagation path of Reg.3 and Reg.4 reflects the process of ground motion in the subduction zone. The attenuation curves of these two regions are between geometric diffusion $R^{-0.5}$ and $R^{-1.0}$, and between geometric diffusion $R^{-0.5}$ and $R^{-1.5}$, respectively. With the increase of hypocentral distance, the attenuation curves show a trend of accelerating decline, indicating that the attenuation rate of the far–field ground motion becomes faster. The path attenuation of ground motion in the subduction zone shows a more obvious frequency correlation, which indicates that there is stronger inelastic attenuation of ground motion in the subduction zone.

The attenuation curves of 0.5 Hz, 1.0 Hz, 2.0 Hz, 5.0 Hz, 10 Hz, and 20 Hz in each region are represented by black, red, cyan, blue, green, and pink, respectively. It is found that in Reg.1 and Reg.2, the attenuation curves of the six frequencies in each region have a consistent downward trend, especially in Reg.2, the attenuation curves of each frequency are almost parallel. However, there is no obvious correlation between attenuation and frequency. The variation of attenuation curves with frequency in Reg.3 and Reg.4 is somewhat similar. The four curves with frequencies less than or equal to 5 Hz are almost distributed in parallel and are negatively correlated with frequency, that is, the higher the frequency, the weaker the attenuation. Among them, the value of the attenuation curve of 5 Hz has the maximum value. The attenuation curve with a frequency of 10 Hz is in line with four attenuation curves (≤5 Hz) when the hypocentral distance is less than ~60 km. With the increase of hypocentral distance, the decrease rate of the attenuation curve is significantly higher than that of the four attenuation curves (≤5 Hz), indicating that the attenuation curve of 10 Hz is obviously different from that of low and medium frequency,

and its attenuation curves decrease more with the increase of hypocentral distance. The attenuation curve of 20 Hz decreases rapidly when the hypocentral distance is greater than ~30 km, and its value is the smallest among the six curves. It can be seen intuitively that the seismic wave weakens obviously with the increase of hypocentral distance.

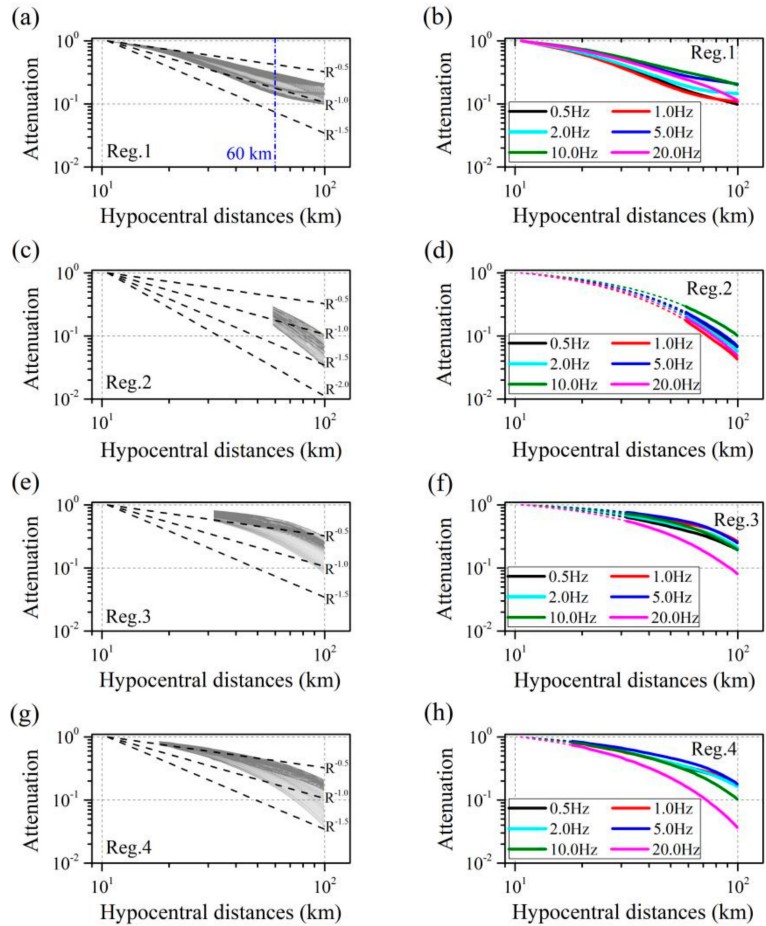

**Figure 7.** 0.25–20 Hz attenuation curves of (**a**) Reg.1, (**c**) Reg.2, (**e**) Reg.3, and (**g**) Reg.4 and comparison with $R^{-0.5}$, $R^{-1.0}$, $R^{-1.5}$ and $R^{-2.0}$ curves. 0.5 Hz, 1.0 Hz, 2.0 Hz, 5.0 Hz, 10 Hz, and 20 Hz attenuation curves of (**b**) Reg.1, (**d**) Reg.2, (**f**) Reg.3, and (**h**) Reg.4.

Concurrently, for analyzing the difference of attenuation curves of the same frequency in the four regions, we compared the attenuation curves of six frequencies (0.5 Hz, 1.0 Hz, 2.0 Hz, 5.0 Hz, 10.0 Hz, 20.0 Hz), as shown in Figure 8. The black, green, blue, and pink solid lines in the figure are the attenuation curves of Reg.1, Reg.2, Reg.3, and Reg.4 at corresponding frequencies respectively. It can be found that at low and medium frequency bands (0.5 Hz, 1.0 Hz, 2.0 Hz, 5.0 Hz), the attenuation curves of Reg.1 and Reg.2 decrease more quickly than those of Reg.3 and Reg.4. All earthquakes in Reg.1 and Reg.2 occurred on land. In Reg.1, there are shallow crustal earthquakes with a focal depth of less than 20 km, and in Reg.2, there are upper mantle earthquakes with a focal depth of more than 40 km. The attenuation curves of the two land regions have little difference within the hypocentral distance of about 60 km. When the hypocentral distance is greater than 60 km, the attenuation curves of Reg.1 decline slowly at a constant speed to almost no attenuation, while that of Reg.2 decline faster and with a smaller value than that of Reg.1. The earthquakes in Reg.3 and Reg.4 occurred in the northeastern and eastern coast of Kanto Basin, respectively, with the characteristics of the subduction zone. The attenuation curves of the two regions attenuate uniformly and slowly at the hypocentral distance less than 60 km and accelerate when the hypocentral distance exceeds 60 km. The medium of

seismic wave propagation in these two regions is seafloor rock, and the different properties between the rock under the oceans and continents may lead to the attenuation of ground motion in the seafloor that is slower than on land.

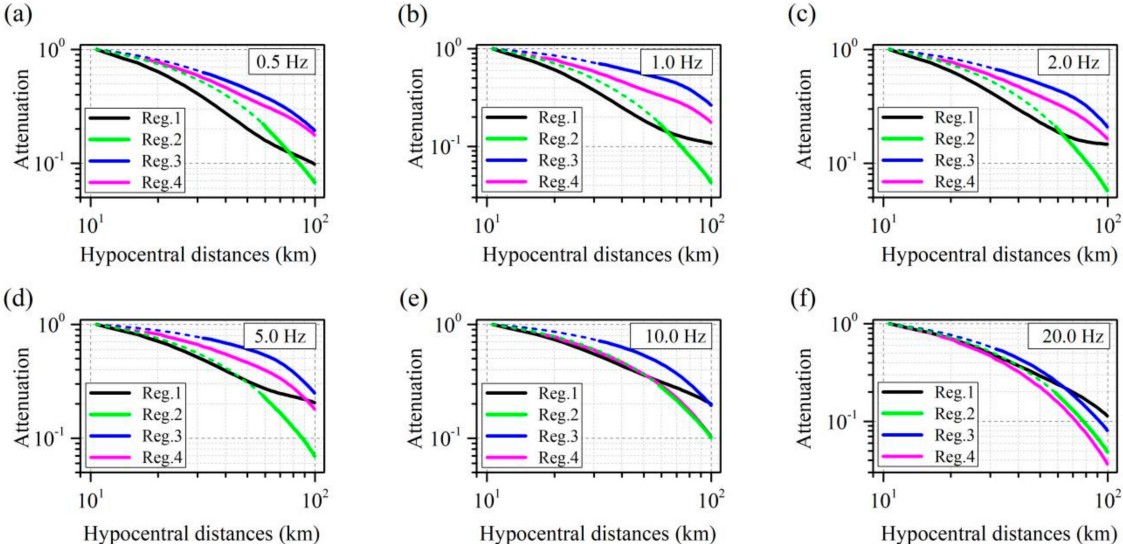

**Figure 8.** Attenuation curves of Reg.1, Reg.2, Reg.3, and Reg.4 for six frequencies, which are (**a**) 0.5 Hz, (**b**) 1.0 Hz, (**c**) 2.0 Hz, (**d**) 5.0 Hz, (**e**) 10 Hz, and (**f**) 20 Hz.

In the high–frequency band (10.0 Hz, 20.0 Hz), the attenuation curves of Reg.3 and Reg.4 gradually decrease faster with the increase of hypocentral distances. At 10.0 Hz, the attenuation curve of Reg.4 almost coincides with that of Reg.2, which has the strongest attenuation in the four regions. Although the value of the attenuation curve of Reg.3 is still the highest, its slope of the curve increases significantly, indicating that the attenuation of Reg.3 is accelerating. At 20.0 Hz, the attenuation curve of Reg.4 declines the most, and that of Reg.3 also drops more than that of Reg.1. Obviously, the attenuation of the ground motion in the subduction zone has been enhanced in the high–frequency band.

By comparing the attenuation curves of ground motion in four regions, it is found that the attenuation of seismic waves is affected by the stratigraphic structure and propagation path. The propagation of seismic waves in the shallow crust (Reg.1) is affected by the refraction of the crust–mantle boundary. Overall, the path attenuation in Reg.1 is weak, with approximately no attenuation over a long distance (>60 km). For the upper mantle earthquakes (Reg.2), the seismic wave travels through the upper mantle and the crust, showing strong attenuation. For earthquakes that occurred in the subduction zone (Reg.3 and Reg.4), the attenuation curve of the attenuation rate increases gradually with the increase of hypocentral distance, and it was faster in the high–frequency band. The frequency correlation of path attenuation is more obvious at a long distance, indicating that the attenuation of earthquakes occurring in the subduction zone contains stronger inelastic attenuation.

The attenuation curves can be simply described by the geometrical spreading and the anelastic attenuation in the function of the S–wave quality factor ($Q$), that is,

$$\ln A(f, R_{ij}) = n \ln(R_{ref}/R_{ij}) - \pi f(R_{ij} - R_{ref})/Q(f)\beta, \tag{5}$$

where $R_{ref} = 10$ km is the reference distance, shear–wave velocity $\beta = 3.6$ km/s, n is the geometrical spreading exponent assumed to be independent of the frequency and distance. The nonlinear least–squares regression was applied to solve Equation (5) for frequency–dependent $Q$ and exponent η. The power exponential form of $Q = Q_0 f^\eta$ is used to represent the frequency–related quality factors, and the $Q$ values of Reg.1, Reg.2, Reg.3, and Reg.4 can be expressed as $92.33f^{1.87}$, $27.75f^{1.08}$, $52.58f^{0.95}$, and $58.07f^{0.89}$, respectively. The $Q$ values

versus frequency are shown in Figure 9. It can be seen that the $Q$ value usually increases with the increase of frequency.

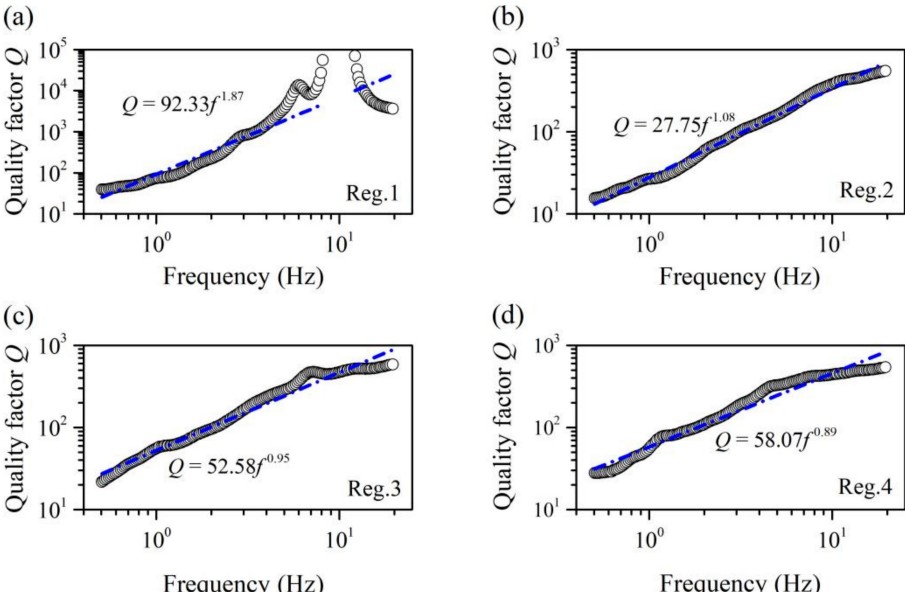

**Figure 9.** Frequency–dependent S–wave quality factor $Q$ and fitting curves of (**a**) Reg.1, (**b**) Reg.2, (**c**) Reg.3, and (**d**) Reg.4. The solid line represents the least squares regression at frequencies of 0.5–20.0 Hz.

The comparison of $Q$ value fitting curves of the four regions is shown in Figure 10. Reg.1 has the highest $Q$ value and Reg.2 has the lowest one. The $Q$ values of Reg.3 and Reg.4 are almost the same, as well as between Reg.1 and Reg.2. In the low–frequency band, the $Q$ values of Reg.1, Reg.2, and Reg.3 are relatively close, which is consistent with the results obtained by Zhao et al. [25] that the attenuation rates of shallow crustal earthquakes and upper mantle earthquakes are in a long period (>2 S). In the high–frequency band (>10 Hz), the $Q$ values of Reg.2, Reg.3, and Reg.4 are also close, indicating that the attenuation curves of Reg.2, Reg.3, and Reg.4 in the high–frequency band are relatively similar. Zhao et al. [25] pointed out that the $Q$ value of the upper mantle of Japan is generally lower than that of many shallow crusts of Japan, which is consistent with the results obtained in this paper.

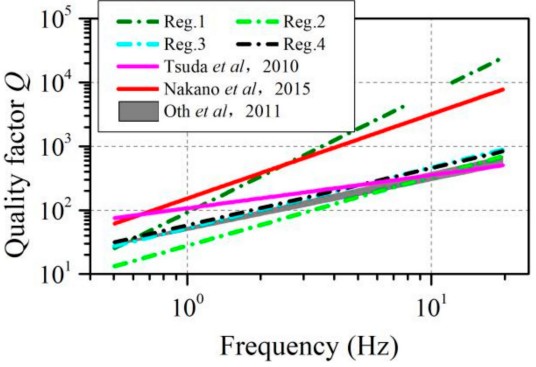

**Figure 10.** Comparison of $Q$ value fitting curves between this study and other studies.

Tsuda et al. [50] studied the attenuation characteristics within the Kanto Basin by using 19 earthquakes that occurred near the basin and found that the S–wave frequency–dependent $Q$ obtained within the research area was $107.33f^{0.52}$ (pink line in Figure 10), which is close to the $Q$ values of Reg.2, Reg.3, and Reg.4 in this study in the medium and high–frequency bands. Oth et al. [34] divided Japan into five independent research regions, among which the fourth region has a similar research scope with the S–wave frequency–

dependent $Q$ of shallow crustal earthquakes in this region being $(51 \pm 3)f^{(0.82 \pm 0.04)}$ (gray shaded area in Figure 10). It is almost consistent with $Q$ in Reg.3 and Reg.4 in the frequency range of 0.5–20 Hz. Nakano et al. [51] not only divided Japan into 6 regions but also divided the earthquakes in each region into three types: shallow crustal, subduction interface, and subduction slab earthquake. Among them, the quality factor of shallow crustal earthquakes in the second region, which is close to the Kanto Basin and its vicinity in this study, is $152.5f^{1.32}$ (red line in Figure 10).

## 5. Conclusions

In this study, according to the focal mechanism and Slab1.0 model, the Kanto Basin and its surrounding areas with complex seismic structures are divided into four regions. A one-step non-parametric generalized inversion technique was used to analyze the seismic wave attenuation characteristics of each region separately. By comparing the attenuation results of the four regions, the conclusions are as follows:

(1) The epicenter of the earthquake in Reg.1 is located in the shallow crust on the northeast land of the Kanto Basin. The seismic wave path attenuation in Reg.1 is relatively wake, and there is approximately no attenuation over a long distance (>60 km), which may be related to the seismic wave refraction at the crust–mantle boundary. The frequency–dependent inelastic attenuation is 0.5–20 Hz, i.e., quality factor $Q = 92.33f^{1.87}$.

(2) The epicenter of the earthquake in Reg.2 is located in the upper mantle in the Kanto Basin. Seismic waves generated by earthquakes in Reg2 travel through the upper mantle and crust as they are received by strong–motion stations, traveling a more complex and longer path than Reg.1, resulting in a stronger attenuation. The slope of the path attenuation curve is approximately parallel to $R^{-2.0}$, and the small quality factor $Q = 27.75f^{1.08}$.

(3) Reg.3 and Reg.4 are located on the eastern coast of the Kanto Basin, and the earthquake types in the regions are complex, with typical subduction zone characteristics. The attenuation of seismic waves in the subduction zone has an obvious frequency correlation, that is, the higher the frequency, the faster the attenuation, indicating that the seismic attenuation of the subduction zone contains strong inelastic attenuation. The frequency–dependent inelastic attenuation $Q$ of Reg.3 and Reg.4 are $52.58f^{0.95}$ and $58.07f^{0.89}$, respectively.

The conclusions show the regional characteristics of attenuation. For land, the attenuation of earthquakes occurring in theshallow crust and upper mantle show an obvious correlation with distance, indicating that the increase of hypocentral distance is conducive to reducing seismic damage. For earthquakes occurring in the subduction zone, the attenuation is correlated with the frequency, and the low–frequency component is the main damage to the surface. This study provides a reference for the attenuation research of the complex plate region in the subduction zone and is of great significance to improve the ability of seismic monitoring and disaster reduction around the Kanto Basin.

**Author Contributions:** Conceptualization, Y.Z. and H.W.; methodology, Y.Z. and T.M.; software, T.M., J.Y.; formal analysis, Y.Z., T.M., J.Y., X.W. and W.Z.; investigation, Y.Z., T.M., H.W., X.W. and W.Z.; resources, X.W. and W.Z.; data curation, Y.Z. and T.M. All authors have read and agreed to the published version of the manuscript.

**Funding:** This research was funded by the Natural Science Foundation of China (Funding No. U1901602, Funder: Dr.Hongwei Wang), Key R&D Project of the Department of Science and Technology of Jilin Province (Funding No. 20210203145SF, Funder: Prof.Xiuli Wang).

**Institutional Review Board Statement:** Not applicable.

**Informed Consent Statement:** Not applicable.

**Data Availability Statement:** The Strong–motion data at strong–motion stations in this study were derived from the National Research Institute for Earth Science and Disaster Prevention (NIED) in Japan. Data can be obtained from the K–NET and KiK–net web sites at www.k-net.bosai.go.jp and www.kik.bosai.go.jp, respectively. The seismic focal mechanism solution automatically measured by the AQUA system of NIED, Japan (https://www.hinet.bosai.go.jp/AQUA/aqua_catalogue.php) (accessed on 21 December 2019), Moment magnitude $M_w$ in the Global Centroid Moment Tensor catalogue was obtained from https://www.globalcmt.org/CMTsearch.html (accessed on 21 December 2019). The $\beta$ was derived from the CRUST 1.0 model available at https://igppweb.ucsd.edu/~gabi/crust1.html (accessed on 21 December 2019). Some of the plots were produced using Generic Mapping Tools.

**Acknowledgments:** The authors gratefully acknowledge the financial support provided by the Natural Science Foundation of China (No. U1901602), Key R&D Project of the Department of Science and Technology of Jilin Province (No. 20210203145SF).

**Conflicts of Interest:** The authors acknowledge there are no conflict of interest recorded.

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
