# Peer review of "Seismic Wave Attenuation Characteristics from the Ground Motion Spectral Analysis around the Kanto Basin"

_buildings, doi:10.3390/buildings12030318_

Round 1

Reviewer 1 Report

The paper "Seismic wave attenuation characteristics from the ground motion spectral analysis around Kanto Basin" reports an interesting work about the evaluation of the seismic wave attenuation characteristics of complex plate tectonics in and around the Kanto Basin. The topic of the manuscript is current and of great interest to the scientific community. The method used in the study is clearly described in the text and the results are discussed comprehensively. For these reasons it is opinion of this reviewer that the manuscript can be considered for the publication in Buildings Journal after the following minor improvements:

  • in "Cosine tapers were added on both ends of the extracted S-waves to eliminate truncation errors, and the length of each taper corresponded to 10% of the S-wave duration[33-34]", consider as reported in Zucca M, Crespi P, Tropeano G, Simoncelli M. "On the Influence of Shallow Underground Structures in the Evaluation of the Seismic Signals", Ingegneria Sismica, 38(1), pp 23-35, 2021
  • improve the quality of the Figure 1, in particular of the legend 
  • in line 56 "Shallow" and line 125 "We" remove the uppercase letter
  • in Section 5 better highlight the original aspects of the work 

Author Response

Responses to Reviewer #1

1.1. Comment: in "Cosine tapers were added on both ends of the extracted S-waves to eliminate truncation errors, and the length of each taper corresponded to 10% of the S-wave duration[33-34]", consider as reported in Zucca M, Crespi P, Tropeano G, Simoncelli M. "On the Influence of Shallow Underground Structures in the Evaluation of the Seismic Signals", Ingegneria Sismica, 38(1), pp 23-35, 2021

Response: We really appreciate the reviewer for providing this useful paper. We carefully read this and much more papers and further revised thoroughly the introduction part of the manuscript. The "On the Influence of Shallow Underground Structures in the Evaluation of the Seismic Signals" was added in the introduction of the revised manuscript (line:177-180).

1.2. Comment: improve the quality of the Figure 1, in particular of the legend 

Response: We sincerely appreciate the reviewer for pointing out this problem. We have modified Figure 1 in the revised manuscript.

1.3. Comment: in line 56 "Shallow" and line 125 "We" remove the uppercase letter

Response: We sincerely appreciate the reviewer for pointing out this mistake, we have modified it in the revised manuscript.  

1.4. Comment: in Section 5 better highlight the original aspects of the work 

Response: We sincerely thank the reviewer for this suggestion. In the conclusion, we added the significance of the research and the summary of the regional attenuation characteristics. The  details are as follows, and the corresponding modifications were made in the revised manuscript (line:399-407). The conclusions show the regional characteristics of attenuation. For land, the attenuation of earthquakes occurring in shallow crust and upper mantle show obvious correlation with distance, indicating that the increase of hypocentral distance is conducive to reducing seismic damage. For earthquakes occurring in the subduction zone, the attenuation is obviously correlated with the frequency, and the low-frequency component is the main damage to the surface. This study provides a reference for the attenuation research of the complex plat region in subduction zone, and is of great significance to improve the ability of seismic monitoring and disaster reduction around the Kanto Basin.

Reviewer 2 Report

In the article, resolution and damping characteristics due to ground motion are explained. However, it is seen that the study is mainly carried out depending on the accelerometer. This subject must be clearly defined.

In text [33-34], Following paper should be added.

AydınBüyüksaraç, Tolga Bekler,  Alper Demirci, Onur Eyisüren New insights into the dynamic characteristics of alluvial media under the earthquake prone area: a case study for the Çanakkale city

settlement (NW of Turkey).  Arabian Journal of Geosciences      (2021) 14:2086

The devastating Mw7.9 must be Mw=7.9

Similar studies can be added to this study in the world not only studied area.

At the end of the introduction, a paragraph should indicate what has been done in the study step by step. It would be beneficial to reveal the novelty of the study more clearly, here.

Adding descriptions and figures with the studied area will attract the reader's attention. The four regions considered in the study should be shown on a more specific map. It should be stated how the zoning was chosen.

In some Figures, the resolution is poor. Horizontal and vertical axes and legends of all figures should be made more specific.

 Earthquake magnitudes and numbers in each region can be given in a table. (By classifying the magnitudes)

Also, an explanation with faults in the region under consideration and adding a figure if possible will enrich the article.

“Finally, a total of 12198 strong-motion recordings were assembled in the dataset that was used for the inversion, with Reg.1 containing 4697 records from 98 earthquakes, Reg.2  containing 2990 records from 69 earthquakes, Reg.3 containing 2412 records from 136  earthquakes, and Reg.4 containing 2109 records from 94 earthquakes.” Values here can be given with the help of a graph.

For Figure 1, earthquake magnitudes can also be coloured according to the figure on the scale part.

After preliminary manual screening, We. Here We must be we.

It would be nice if Figure 4 could make it a little more specific if possible.

In Figure for in legend crastal is crustal

In reference 38 must be checked by authors.

Author Response

Responses to Reviewer #2

2.1. Comment: In text [33-34], Following paper should be added. AydınBüyüksaraç, Tolga Bekler, Alper Demirci, Onur Eyisüren New insights into the dynamic characteristics of alluvial media under the earthquake prone area: a case study for the Çanakkale city settlement (NW of Turkey).  Arabian Journal of Geosciences (2021) 14:2086

Response: We really appreciate the reviewer for providing this useful paper. We carefully read this and much more papers and further revised thoroughly the introduction part of the manuscript. The New insights into the dynamic characteristics of alluvial media under the earthquake prone area: a case study for the Çanakkale city settlement (NW of Turkey) was added in the introduction of the revised manuscript.

2.2. Comment: The devastating Mw7.9 must be Mw=7.9.

Response: According to the reviewer’s suggestion, the Mw7.9 was replaced by Mw=7.9 in the revised manuscript.

2.3. Comment: Similar studies can be added to this study in the world not only studied area.

Response: Following this suggestion, we added the development and application of methodology to the introduction. The corresponding modifications were made in the revised manuscript (line:64-80).

2.4. Comment: At the end of the introduction, a paragraph should indicate what has been done in the study step by step. It would be beneficial to reveal the novelty of the study more clearly, here.

Response: According to the reviewer’s suggestion, in the last paragraph of the introduction, we describe in detail the steps of the research as well as the innovation and significance of the research (line:81-94).

2.5. Comment: Adding descriptions and figures with the studied area will attract the reader's attention. The four regions considered in the study should be shown on a more specific map. It should be stated how the zoning was chosen.

Response: We are very grateful to the reviewers for this valuable comment, which makes us realize that the description of regional division in the previous manuscript is not detailed enough. We describe in detail the reasons for the initial rough division of regions, and emphasize that the region covered by the final data set after data processing is the specific research scope of each region (line:132-149,211-223). Concurrently, we also provide the distribution of the four regions according to the suggestions, as shown in Figure 5 and Table 2 in the revised manuscript.

2.6. Comment: In some Figures, the resolution is poor. Horizontal and vertical axes and legends of all figures should be made more specific.

Response: We sincerely appreciate the reviewer for pointing out this problem. In order to prevent the image resolution from being reduced during the upload process, I uploaded the 300dpi image of each figure output by Origin software separately. In addition, according to the reviewer's suggestion, Figure 2, Figure 7, Figure 8, Figure 9 and Figure 10 in the revised manuscript were modified to add a caption on each horizontal and vertical axis.

2.7. Comment: Earthquake magnitudes and numbers in each region can be given in a table. (By classifying the magnitudes)

Response: According to the reviewer’s suggestion, the number of recordings and earthquakes, magnitude, focal depth, Hypocentral distance in each region are shown in Table 2. the details are as follow, and the corresponding modifications were made in the revised manuscript.

Table 2. Number of recordings and earthquakes, magnitude, focal depth, Hypocentral distance in each region

Region

Number of recordings

Number of earthquakes

Magnitude

Focal depth

Hypocentral distance

Reg.1

4697

98

MJMA 4-5.4

4-17 km

10-100 km

Reg.2

2990

69

MJMA 4-5.5

41-67 km

59-100 km

Reg.3

2412

136

MJMA 4-6.2

13-55 km

10-100 km

Reg.4

2109

94

MJMA 4-6.1

10-43 km

18-100 km

2.8. Comment: Also, an explanation with faults in the region under consideration and adding a figure if possible will enrich the article.

Response: We sincerely thank the reviewer for this suggestion. Actually, we have tried to consider the representation of fault information in the base map. However, since the scope of our study area is small and far from the main fault, and our research content has little correlation with the main fault. If we add the main fault, the resolution of the study area will be reduced.

2.9. Comment:“Finally, a total of 12198 strong-motion recordings were assembled in the dataset that was used for the inversion, with Reg.1 containing 4697 records from 98 earthquakes, Reg.2  containing 2990 records from 69 earthquakes, Reg.3 containing 2412 records from 136  earthquakes, and Reg.4 containing 2109 records from 94 earthquakes.” Values here can be given with the help of a graph.

Response: According to the reviewer’s suggestion, we made modifications and presented the information in Figure 6 and Table 2 of the revised manuscript.

2.10. Comment: For Figure 1, earthquake magnitudes can also be coloured according to the figure on the scale part.

Response: According to the reviewer’s suggestion, we drew a map with scale as the color parameter, shown in Figure S1. Unfortunately, we found that although this is more conducive to distinguishing the scale, it can not reflect the focal depth. When we first divided the earthquake types, we used Slab1.0, the focal depth is an important parameter. Moreover, using the focal depth as the color parameter is more clear for viewing the information such as focal distribution regionalization. To understand the fundamental problem and interest of our research more intuitively, we think that take depth as a color parameter will be better.

Figure S1 The location, magnitude and focal depth of 881 earthquakes that occurred in the range of 139-141.5°E, 35-37.5°N from January 1, 2010 to December 31, 2019.

2.11. Comment: After preliminary manual screening, We. Here We must be we.

Response: We sincerely appreciate the reviewer for pointing out this mistake, we have modified it in the revised manuscript.  

2.12. Comment: It would be nice if Figure 4 could make it a little more specific if possible.

Response: We sincerely appreciate the reviewer for pointing out this problem. We have modified Figure 4, and we also provide the distribution of the four regions according to the suggestions, as shown in Figure 5 in the revised manuscript.

2.13. Comment: in Figure 4 in legend crastal is crustal.

Response: We sincerely appreciate the reviewer for pointing out this mistake, we have modified it in the revised manuscript.

2.14. Comment: In reference 38 must be checked by authors.

Response: We sincerely appreciate the reviewer for pointing out this mistake, we have modified it in the revised manuscript.
